# A Computer Vision System Based on Majority-Voting Ensemble Neural Network for the Automatic Classification of Three Chickpea Varieties

**DOI:** 10.3390/foods9020113

**Published:** 2020-01-21

**Authors:** Razieh Pourdarbani, Sajad Sabzi, Davood Kalantari, José Luis Hernández-Hernández, Juan Ignacio Arribas

**Affiliations:** 1Department of Biosystems Engineering, College of Agriculture, University of Mohaghegh Ardabili, Ardabil 56199-11367, Iran; r_pourdarbani@uma.ac.ir (R.P.); sajadsabzi2@gmail.com (S.S.); 2Department of Mechanics of Biosystems Engineering, Faculty of Agricultural Engineering, Sari Agricultural Sciences and Natural Resources University, Sari 48181 68984, Iran; d.kalantari@sanru.ac.ir; 3Division of Research and Graduate Studies, TecNM/Technological Institute of Chilpancingo, Chilpancingo 39070, Mexico; joseluis.hernandez@itchilpancingo.edu.mx; 4Department of Teoría de la Señal y Comunicaciones e Ingeniería Telemática, University of Valladolid, 47011 Valladolid, Spain; 5Castilla-León Neuroscience Institute (INCYL), University of Salamanca, 37007 Salamanca, Spain

**Keywords:** *Cicer arietinum* L., chickpea, classification, computer vision, feature selection, hybrid ANN, image processing, legume, machine learning, majority voting, segmentation

## Abstract

Since different varieties of crops have specific applications, it is therefore important to properly identify each cultivar, in order to avoid fake varieties being sold as genuine, i.e., fraud. Despite that properly trained human experts might accurately identify and classify crop varieties, computer vision systems are needed since conditions such as fatigue, reproducibility, and so on, can influence the expert’s judgment and assessment. Chickpea (*Cicer arietinum* L.) is an important legume at the world-level and has several varieties. Three chickpea varieties with a rather similar visual appearance were studied here: Adel, Arman, and Azad chickpeas. The purpose of this paper is to present a computer vision system for the automatic classification of those chickpea varieties. First, segmentation was performed using an Hue Saturation Intensity (HSI) color space threshold. Next, color and textural (from the gray level co-occurrence matrix, GLCM) properties (features) were extracted from the chickpea sample images. Then, using the hybrid artificial neural network-cultural algorithm (ANN-CA), the sub-optimal combination of the five most effective properties (mean of the RGB color space components, mean of the HSI color space components, entropy of GLCM matrix at 90°, standard deviation of GLCM matrix at 0°, and mean third component in YCbCr color space) were selected as discriminant features. Finally, an ANN-PSO/ACO/HS majority voting (MV) ensemble methodology merging three different classifier outputs, namely the hybrid artificial neural network-particle swarm optimization (ANN-PSO), hybrid artificial neural network-ant colony optimization (ANN-ACO), and hybrid artificial neural network-harmonic search (ANN-HS), was used. Results showed that the ensemble ANN-PSO/ACO/HS-MV classifier approach reached an average classification accuracy of 99.10 ± 0.75% over the test set, after averaging 1000 random iterations.

## 1. Introduction

In 2017, global production of chickpea (*Cicer arietinum* L.) was estimated in almost 15 million tons/year. In addition, the average yield of chickpea in Iran is lower than the world average. This difference between the mean and potential yield of this product may be due to its susceptibility to biotic and non-biotic constraints [1]. About 75% of chickpea is used for human consumption and from the remaining, 14% is used for livestock consumption, 7% is used for seed production, and 4% is wasted at different stages of processing [2]. The importance of chickpea grading is highlighted when consumers go to different distributors and ask them to prepare their order. In other words, different varieties of chickpea have different applications, properties and nutritional values. Therefore, it is important to correctly identify the product variety. This assessment is often made by human experts, but for various reasons, such as expert psychological changes, fatigue, and so on, errors in the evaluation of different varieties may occur. For this reason, it might be important to provide rapid detection systems using new technology. One can consider at least two main reasons why it might be important to provide rapid detection systems using new technology: first, different applications and price of each chickpea variety; second, separation of each chickpea variety when stored in the same tank, once they have been harvested. Different methods have been proposed for the grading of different products. Some of these systems are based on physical and mechanical properties [3,4,5]. Although these systems are simple and inexpensive, they are not popular because of their often low efficiency and capacity. For instance, in order to separate ripe fruit from immature, fruits were fed in a row on a v-shaped conveyor. At the sorting points, the piston behind the belt vibrated at a predetermined resonance frequency, producing movement over the fruits. Vibrant immature fruits were cut into the lateral conveyor belts and separated from ripened ones. Another example was the use of elasticity feature in the sorting of ripe fruit. Fruits were given a free fall onto a rotating cylinder. As the rotary cylinder moved, the fruits had a specific downward parabola motion. Ripe fruit with soft texture would be directed to closer outlet and unripe fruit with firm texture would be directed farther to the second outlet. Since these classical food separation systems have a high response time as compared to new computer vision techniques, thus classical system capacity is low. In addition, it is also possible to damage fruit piece texture due to free fall or mechanical vibration [6]. In recent years, the application of new technologies such as computer vision and machine learning has been increasing in various areas of the agricultural and food industries. These systems play an important role in the precise and uniform control of food quality by eliminating manual inspection and providing an appropriate objective, automatic, and non-intrusive way to distinguish the desired product characteristics (color, size, shape, texture, and pests) for separation and classification [7,8] purposes. Iannace et al. developed a model based on the artificial neural network to detect unbalanced blades in an unmanned aerial vehicle (UAV) propeller. An air collision or a defect in the propellers of a drone can cause the aircraft to fall to the ground, and thus its consequent destruction [9]. LeCun et al. recognized document content using a gradient-based learning rule [10]. Uijlings et al. introduce a selective search which combines the strength of both an exhaustive search and segmentation for object recognition that results in a small set of data-driven, class-independent, high quality locations, yielding a 99% recall value [11]. Kurtulmus and Unal [12] studied on seven different varieties of rapeseed. Mass level imaging method was selected instead of kernel surface imaging for rapeseed classification. Mass surfaces were constituted by spreading rapeseed samples in a 15–20 mm layer in a square frame in order to prevent the scanner light from passing through the mass layer. Before scanning of each individual sample, the scanner glass was thoroughly cleaned. An imaging chamber was designed using a charged-coupled device (CCD) scanner and a personal computer under laboratory-controlled conditions. Samples were placed in a frame of 36 × 36 mm^2^ to obtain a uniform image. A total of 525 samples were obtained by scanning 75 rapeseed samples from each variety in lab. Three classes of texture properties were extracted from rapeseed image samples: Gray Level Co-ocurrence Matrix (GLCM), defined as a space frequency matrix between adjacent pixels in a digitized image, Gray Level Run Length Matrix (GLRLM), defined as the number of runs with pixels of gray level *i* and run length *j* at a given image direction, and local binary patterns, that transform an image into an array of integer labels describing small-scale appearance on an image. Next, three classifiers including support vector machine classifications, k-nearest neighbor, and stochastic gradient descent, were used to classify seven rapeseed varieties. Results showed that the classifiers clustered samples with good accuracy. Other researchers have proposed a new approach based on image analysis techniques to discriminate defective lentil seeds from healthy ones. They extracted several color, shape and texture features. These features were then used as inputs for Support Vector Machine (SVM) classification. Results showed that their method achieved accuracy was 98.9% [13].

Liu et al. [14] identified different varieties of soybean as well as healthy versus unhealthy seeds using image processing. For this purpose, 857 soybeans were collected, and then imaging was performed. Shape, texture, and color characteristics were extracted. Finally, classification was performed using an artificial neural network classifier with a 97% accuracy. Golpour et al. [15] used texture features of digital images to identify paddy, brown, and white rice varieties. In this study, 2.5 kg with 11–12% moisture content of each variety was prepared. A total of 90 3D images of each variety were captured under standard conditions. A total of 1350 images of 540 × 390 pixels were generated. The classification was performed with a 97.67% accuracy.

The aim of the present paper is to develop a computer vision system in order to classify three similar varieties of chickpea (Adel, Azad, and Arman) using an image processing and machine learning ensemble ANN majority voting (MV) approach between hybrid artificial neural network- particle swarm optimization (ANN-PSO), hybrid artificial neural network-ant colony optimization (ANN-ACO) and hybrid artificial neural network-harmonic search (ANN-HS) outputs.

## 2. Materials and Methods

In this study, in order to train a computer vision system to be able to identify different chickpea varieties, different steps are required. Figure 1 illustrates a flowchart for the various stages needed for the proper design and functioning of our system. As can be seen in Figure 1, five main blocks exist in our chickpea automatic classification system, being each block explained as follows.

### 2.1. Varieties of Chickpea Used

Three different chickpea varieties, that are similar in appearance, were used to train the computer vision system: Adel, Arman, and Azad. Adel variety is used mainly for making cakes and muffins in the food industry, while Arman variety is used for livestock consumption, and Azad variety is used as powder in various food preparations. Chickpeas are spread randomly, are not isolated and they might touch one to each other. Each image contains around 60 chickpea samples. After each capture, chickpeas were removed and replaced with a different batch sample. Samples were collected from chickpea fields in Kermanshah, Iran (longitude 7.03° E; latitude 4.22° N). Figure 2 shows an example image of each chickpea variety. 196, 192, and 197 images were obtained from Adel, Arman, and Azad chickpea varieties, respectively. Images were taken using an industrial camera DFK 23GM021 (The Imaging Source GmbH, Bremen, Germany), which has a 1/3 inch CMOS sensor, located at a fixed distance of 15 cm over the samples. To create uniform light imaging conditions, white LEDs with 425 lux intensity were used.

### 2.2. Segmentation Operation

As is well-known, a color space is an input space including three main color components. Each color space presents a particular value of a {3×1} vector size value for each pixel, comprising each of the three scalar color components in that particular color space. Since the values of pixels in background and chickpeas are different, it might be possible to separate pixels in chickpea (foreground) from pixels in the background.

Hue Saturation Intensity (HSI) color space was used to segment chickpeas from background. In fact, after examining different color spaces and applying different thresholds, this color space was selected. Equation (1) describes segmentation threshold. This equation indicates that if the pixel with the first *Hue* channel of the color space is less than 0.4 or the second *Saturation* channel of this color space is greater than 0.15 or the third channel *Intensity* of this color space is greater than 0.07, then the pixel belongs to chickpea (foreground), otherwise pixel is part of background.
(1)if either H<0.4  OR  S>0.15 OR  I>0.07 then foreground (chickpea), else background

Figure 3 shows a flowchart to better understand the meaning of segmentation Equation (1).

In Figure 4, a sample image through the whole segmentation process is shown. As a first step, the original image is converted to HSI color space. In the second step, based on Equation (1), most background pixels are removed. Finally, the remaining background pixels are removed based on morphology operator, removing objects with a number of pixels of less than 100. The last value was selected after trial and error.

### 2.3. Extraction of Different Properties of Each Chickpea Sample Image

After segmentation, different color and texture properties (features) of the chickpea samples were extracted. The extracted color properties follow next: mean and standard deviation of the first, second, third channels (components), and the mean of all three channels for RGB, HSV, HSI, YCbCr, CMY, and YIQ color spaces [16]. Table 1 defines the six color spaces used in the present study, including proper transformation equations from RGB color space.

Since there are two properties, four channels (three color components plus the average of all them three) and six color spaces, the total number of extractable properties in this case equals 2×4×6=48. Texture properties from the input samples are related to the GLCM matrix. These properties were extracted for four different neighborhood angles, including 0, 45, 90, and 135 degrees. Table 2 shows the 20 extracted texture properties of each sample. Since there are 20 texture properties and four possible neighborhood angles, the total extractable GLCM texture properties are 20×4=80.

### 2.4. Feature Selection

Since the purpose of the proposed algorithm is to identify different chickpea varieties on-line, the computation time is a critical parameter. For this reason, it is not possible to use all extracted properties because of their time-consuming calculations in the classifications, together with the fact of potential over-learning and thus poor generalization capability to the disjoint test set of an excessive number of input features resulting in an excessive complex network. Therefore, it is necessary to select effective properties among all extracted properties. In this paper, a hybrid artificial neural network-cultural algorithm (ANN-CA) was used to select effective discriminant properties. CA is an optimization algorithm that is inspired by cultural evolution and the impact of cultural and social space. In a society, everyone who is most well- known has the most direct and indirect influence on cultural evolution. In fact, these people will influence how they live, talk, walk, dress, and so on. In fact, the ultimate goal of this algorithm is to find and develop these elites for cultural evolution [18]. In order to select effective properties, this algorithm first considers the whole properties as a vector and then generates different vectors of different smaller sizes and sends them as input to the artificial neural network. In order to select effective properties, this algorithm first considers the whole properties as a vector and then generates different vectors of different smaller sizes such as 3, 5, 9, and 11, and sends them as input to the artificial neural network. Data samples were randomly divided into three disjoint sets. The first set contains 70% of the data samples for training, the second one contains 15% of the data for validation and the third one contains 15% of the data for testing, being all sets disjoint. In fact, the input of the artificial neural network contains the extracted properties selected by CA algorithm and its outputs are the class of the different chickpea varieties to whom the sample is expected to belong to. The mean squared error (MSE) of each vector that enters the ANN is recorded. Finally, any feature vector with the lowest MSE is selected as the optimal vector, and the properties within that vector are selected as the effective discriminant properties. Table 3 gives the parameters of the neural network used to select effective properties with CA algorithm.

### 2.5. Ensemble Classification of Different Chickpea Varieties: Majority-Voting (MV)

Since the main purpose of the ANN majority-voting (MV) method [20] is to perform a merged (combined) classification of three chickpea cultivars, three hybrid neural network classifiers were performed: ANN-PSO, ANN-ACO and ANN-HS. MV decision is based on voting from each independent classifier and agreeing as final consensus winning output the class which was voted most among the various single classifiers.

#### 2.5.1. Hybrid ANN-PSO Classifier

The multilayer perceptron (MLP) ANN has several adjustable parameters that optimally adjust to ensure its high performance [21]. These adjustable parameters include: the number of neurons, the number of layers, the transfer function, the back-propagation network training function, and the back-propagation weight/bias learning function. The number of neurons per layer ranged between 0 and 25 (first layer being between 1 and 25 neurons). The number of layers is at least 1 and at most 3. Transfer functions that can be selected including: *tansig, logsig, purelin, hardlim, compet, hardlims, netinv, poslin, radbas, satlin, satlins, softmax, tribas*. The backpropagation training algorithm will be selected from: *trainlm, trainbfg, trainrp, traincgb, traincgf, traincgp, traincgb, trainscg, trainoss, traingda, traingdx, trainb, trainbfgc, trainbr, trainbuwb, trainc, traingdm, trainr, trains*. Finally, the weight/bias learning function can be selected among functions: *learngdm, learngd, learncon, learnh, learnhd, learnis, learnk, learnlv1, learnlv2, learnos, learnp, learnpn, learnsom, learnsomb, learnwh*. These abbreviations are functions defined under MatlLab software, [19]. Here, particle swarm optimization (PSO) was used in order to adjust parameter values. PSO is a meta-heuristic algorithm that mimics the mass movements of birds/bees to optimize different problems. This algorithm was first proposed by Kennedy and Eberhart [22]. Each answer to the problem is considered a bit. Every single particle is constantly searching and moving. The motion of each particle depends on three factors: the current position of the particle, the best position the particle has ever had, and the best position the entire set of particles has ever had. This algorithm considers the parameters in the form of a vector with a minimum size of 4 and a maximum of 8 to adjust the parameters of the artificial neural network. For example, consider the vector x=[12, 17, tansig, poslin, traingda, learnhd]. This vector shows that the proposed neural network is formed by a two-layer ANN through particle swarm algorithm with 12 and 17 neurons and *tansig* and *poslin* transfer functions. It is also a function of the proposed *traingda* back-propagation network training and *learnhd* back-propagation weight/bias learning function. After adjusting the artificial neural network parameters by this algorithm, the MSE of the ANN output for each input vector selected by the PSO algorithm is stored. This way, the inputs of the ANN are the selected discriminant features and its output are the different classes (chickpea varieties). Finally, any vector with the least MSE is considered as the optimal vector and the values of the parameters in that vector are considered as optimal values. In the present study, more than 2500 different structures of the network were investigated, and an optimal structure was finally selected. After selecting the optimal structure to evaluate its validity, 1000 repetitions (replicates) were performed uniform selecting train, validation, and test set samples. It should be noted that 70% of the data were used for training and validation and 30% for testing.

#### 2.5.2. Hybrid ANN-ACO Classifier

The steps of selecting the ANN- ACO adjustable parameters are exactly like the PSO algorithm used in the hybrid ANN-PSO approach. This algorithm is a method for solving optimization problems. The algorithm is implemented based on ants’ path to find food. In fact, depending on the path length and quality of food, the ants put some amount of substance called pheromone in their path. Other ants also sense the pheromone odor and follow the same path, so the pheromone value in that path increases. The shortest route from the nest to the food source has more pheromones, so the shortest route is chosen by ants [23]. Based on this principle, an ant-community optimization algorithm was implemented in MatLab software and used for the present study.

#### 2.5.3. Hybrid ANN-HS Classifier

In this case, HS algorithm, like the two above mentioned algorithms, adjusts the adjustable parameters of the MLP neural network. This algorithm is also a meta-heuristic algorithm that imitates the natural process of music optimization. In song composition, the gamut of each instrument determines the beauty and harmony of the song when several instruments are played at the same time, in other words, each instrument must be optimized, and therefore the value of the objective function is determined by the values of the variables [24].

#### 2.5.4. Ensemble Final Classification through MV

After performing classification with hybrid ANN-PSO, hybrid ANN-ACO and hybrid ANN-HS algorithms, the MV approach was used to finalize the results combining their outputs. In precise terms, according to the MV method, the output class with the highest vote among the three independent classifiers is selected as the final winning output class.

### 2.6. Optimal Structures of ANNs Adjusted by Different Algorithms

Table 4 gives the optimal structure of the ANN-PSO, ANN-ACO, and ANN-HS hybrid classifiers.

As can be inferred from Table 4, the best structure for all methods is a structure with three layers. Also, in this table it is shown that the best number of neurons of each layer that was selected, based on the ACO, PSO, and HS algorithms, are below 20, thus implying that there is no need for networks to be highly complex.

### 2.7. Criteria Used to Evaluate the Performance of the Different Classifiers: Confusion Matrices and Receiver Operating Curves (ROC) (Test Set)

Two categories of indices were used to evaluate the performance of different classifiers. First group consists in the parameters related to the confusion matrix. This matrix is a square matrix that has the number of classes as number of rows and columns. For example, if there are 3 classes, the matrix has a 3×3 dimension. Rows in this matrix show the actual (true) class value and columns the predicted class value. Parameters that evaluate the performance of the classifiers based on this matrix include sensitivity or recall, accuracy, specificity, precision, and F1-score. In the following, we will define each parameter:

Below are the formulas to be used inside confusion matrix:Sensitivity, recall, true positive (TP) rate or probability of detection: measures the proportion of actual positives that are correctly identified as such (2)Accuracy or correct classification rate (CCR): total percentage of correct system classifications (3)Specificity or true negative (TN) rate: percentage of inaccurate samples that are correctly identified (4)Precision or positive predictive value: is the fraction of relevant instances among the retrieved instances (5)F1-score: recall and precision harmonic weighted average (6).
(2)$$\mathrm{Sensitivity}=\mathrm{Recall}\left(r\right)=\frac{\mathrm{TP}}{\mathrm{TP}+\mathrm{FN}}\times 100$$
(3)$$\;\mathrm{Accuracy}\;\left(\mathrm{CCR}\right)=\frac{\mathrm{TP}+\mathrm{TN}}{\mathrm{TP}+\mathrm{TN}+\mathrm{FP}+\mathrm{FN}}\times 100$$
(4)$$\;\mathrm{Specificity}\;=\frac{\mathrm{TN}}{\mathrm{TN}+\mathrm{FP}}\times 100$$
(5)$$\;\mathrm{Precision}\left(p\right)=\frac{\mathrm{TP}}{\mathrm{TP}+\mathrm{FP}}\times 100$$
(6)$$\;F1-\mathrm{score}=2\frac{\mathrm{pr}}{p+r}$$

Here, TP is equal to the number of samples in each class that are correctly classified. TN is equal to the number of samples on the main diagonal of the confusion matrix minus the number of samples correctly classified in the class in question. False negative (FN) is defined as the sum of the horizontal samples of the class under consideration minus the number of samples that are correctly classified in the class in question. Finally, false positive (FP) is the sum of the vertical samples of the class examined minus the number of samples that are correctly classified in the considered class [25].

Second performance category comprises the ROC chart (plot). This chart has two axes and a bisector straight line: the horizontal axis being the FP rate, $\frac{\mathrm{FP}}{\mathrm{TN}+\mathrm{FP}}$, (1 − specificity), and the vertical axis being the TP rate, sensitivity, c.f. Equation (2). The ROC curve is generated by slowly varying the output detection threshold of the classifier and counting the fraction of FP and TP rates over the test set for each given classifier detection threshold level. The closer the ROC curve goes to the upper-left corner, the high performance of the classification system. Therefore, whenever the diagram has an orthogonal look, this indicates high performance of the classification system [26]. A numerical objective criterion that examines the performance of the classifier in the {1−specificity, sensitivity} ROC chart plane, or alternatively the {FPR,TPR} plane, is the area under the ROC graph (AUC). The minimum value for this criterion is 0.5 (equivalent to the toss of a coin) and the maximum is 1 (perfect classification with 100% accuracy, FP=FN=0).

## 3. Results

### 3.1. Effective Discrimiant Property (Feature) Selection

Discriminant properties selected by the ANN-CA include five features: average channels of first, second and third RGB color space, average first channel of HSI color space, neighborhood entropy of 90° and 0° in GLCM, and the third channel of YCbCr color space.

### 3.2. Classification Using Hybrid ANN-PSO Classifier

Table 5 gives confusion matrix, classification error (per class) and the accuracy (CCR) of the hybrid ANN-PSO classifier, after 1000 iterations. As can be observed, from the 177,000 data samples, only 2385 samples are incorrectly classified in a class other than their original true class, which reduces the maximum possible accuracy of ANN-PSO classification by a factor of only 1.35%.

Figure 5 shows a box diagram of CCR and AUC levels below the ROC curve of Adel, Arman, and Azad classes (AUC1, AUC2, and AUC3, respectively). Among the 1000 replicates, only 10 replicates had a CCR below 95%. Also, more than 50% of replicates have a CCR above 99%. Therefore, it can be concluded that this classifier has been able to be performed the classification with high accuracy.

### 3.3. Classification Using Hybrid ANN-ACO Classifier

Table 6 gives the confusion matrix, error rate and the correct classification rate of the ANN-ACO classifier after averaging 1000 random iterations. As can be seen, the highest classification error belongs to the Adel variety class where from 59,000 input samples, a total of 941 samples are incorrectly classified resulting in a classification error of 1.59%. The lowest classification error belongs to the Azad class, which is less than 0.5%. Interestingly, both the ANN-PSO and ANN-ACO classifiers did not misclassify any single Azad samples as Arman ones, indicating the robustness of the classifiers.

Figure 6 illustrates a box diagram of CCR and AUC under ROC curve of the three classes for ANN-ACO classifier, 1000 random repeated simulations. As can be seen from Figure 6, in this case only two simulations out of 1000 have a correct classification rate below 95%. On the other hand, the boxplot diagrams for the areas under the ROC curve (AUC) of the three chickpea varieties are presented, with values above 0.95 in all 1000 simulations. The more compact box diagram, however, indicates the closer proximity of the results in different iterations, resulting in a high level of classification accuracy with very limited dispersion.

### 3.4. Classification Using Hybrid ANN-HS Classifier

Table 7 gives the confusion matrix, the correct and incorrect classification rates of each class for the ANN-HS classifier case, after performing 1000 independent iterations. The mean correct classification rate of this classifier is 98.99% being able to have the least classification error among the three different classifier techniques implemented.

Figure 7 shows the box diagram of the CCR, and AUC of hybrid ANN-HS after 1000 iterations. It also shows that more than half of the simulations have a correct classification rate over 99%.

## 4. Discussion

Table 8 gives confusion matrix and correct and incorrect classification rates of each class using ANN ensemble PSO/ACO/HS majority-voting (MV) ensemble of neural networks. Among the 177,000 samples, only 1592 samples were incorrectly classified resulting in a correct classification rate of 99.10%. This rate is very high and reflects the superiority of the combined neural network ensemble majority voting approach over other single hybrid ANN classification methods, as expected given the fact that it makes use of information coming from three independent classifiers.

Figure 8 shows a box diagram of the correct classification rate (CCR) and areas under the ROC curve (ACU) of the three classes by the neural ensemble Majority-Voting (MV) after 1000 iterations. As shown in the box diagrams MV CCR is above 99% in more than 50% of the simulations.

Regarding boxplot diagrams for AUC, they also show that more than half of the simulations have values above 0.99. It can be concluded that MV method performs very well in classifying different chickpea varieties.

Table 9 gives the different criteria for evaluating the performance of different classifiers used here after 1000 iterations. All classifiers have accuracy mean values above 98%, implying that all classifiers have high performance in classifying different chickpea varieties. By comparing the accuracy of the classifiers, it can be found that the highest accuracy is for the MV method with a mean value of 99.10%. Also, the MV method has a lower standard deviation value than other classifiers, indicating the robustness of the MV method in accurately classifying all three chickpea varieties. Figure 9 shows the ROC curves by the four hybrid classifiers: ANN-PSO, ANN-ACO, ANN-HS and ANN-PSO/ACO/HS ensemble majority-voting (MV). As can be seen, the ROC curves for each of the three classes produced by the ANN-MV scheme are very close to the top-left chart corner, indicating its superior performance.

We have also computed precision-recall curves AUC as an additional measure of the classifier performance, as shown in Table 10. Consistent results are shown for all four classifiers, but the best pr-AUC for Adel and Arman varieties is found with the ANN-ACO classifier, while the best pr-AUC value for Azad variety is found with the ANN-HS classifier.

Finally, although it is not possible to directly compare the results of the present study due to different seed, different imaging conditions, different image database and other parameters, it may be useful to compare the accuracy of the method here proposed with others from the state of the art. To that aim, Table 11 compares the accuracy of different classifiers used by other authors in classify different legume varieties. The proposed method has a higher number of input samples and higher accuracy as compared to others.

To sum up, the proposed system works in four different stages described as follows:Chickpea bunch imaging acquisition under light controlled conditions, with white LEDs with 425 lux intensity.Automatic chickpea image segmentation.Automatic extraction of different discriminant features, including: average channels of first, second, and third RGB color space, average first channel of HSI color space, neighborhood Entropy of 90° and 0° in GLCM, and third channel of YCbCr color space, from each input sample image.Output chickpea variety classification by a neural network ensemble majority-voting.

## 5. Conclusions

Given that there is always some powder amount in the pile, segmentation should be performed with high accuracy, since if the powder is not properly eliminated during segmentation, results will be produced in the feature extraction phase. After examining different color spaces, HSI was selected as a suitable color space for segmentation purposes.

Among the effective (discriminant) properties selected by the hybrid ANN-CA, there are three color and two texture properties, the last ones belonging to the gray surface co-occurrence matrix. This implies that it is important to identify Adel, Arman, and Azad chickpea varieties using a combination of both color and texture properties.

The evaluation of classifier results with multiple iterations is more realistic than only single run due to the random nature of neural networks learning phase. For that reason, after determining the best structure of neural networks, 1000 iterations were performed to obtain statistically valid results.

The combined majority voting ensemble provides more confident results since it is based on the classification of the majority of three different hybrid ANN classifiers. The results showed that majority voting had the highest classification accuracy and lower dispersion (variance) as compared to the single hybrid ANN classifiers. Given the high accuracy of the proposed approach, we believe it could in principle be extended and applied under real conditions in the food industry in order to automatically classify several chickpea varieties. The input digitized chickpea image database is included as a 37 s mp4 video file for reproducibility and comparison purposes as Appendix A.

## Figures and Tables

**Figure 1 foods-09-00113-f001:**
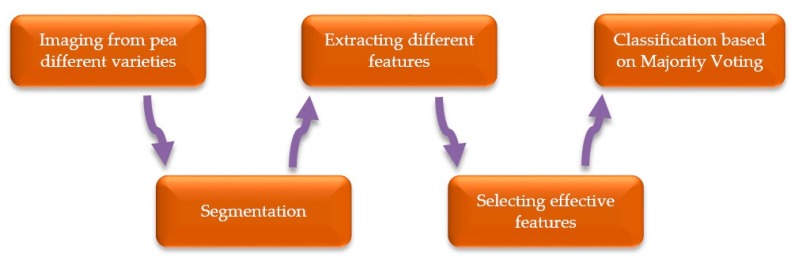
System block diagram for the classification of different chickpea (*Cicer arietinum* L.) varieties.

**Figure 2 foods-09-00113-f002:**
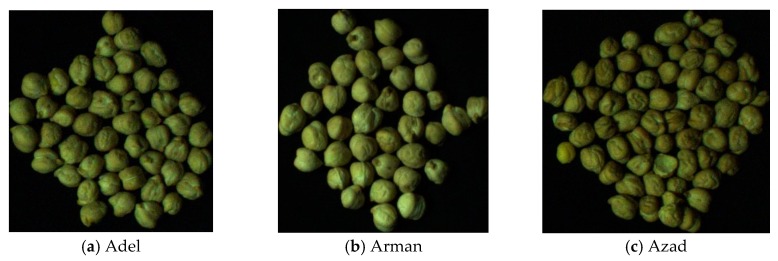
A sample image of each of the three chickpea (*Cicer arietinum* L.) varieties under consideration: (**a**) Adel, (**b**) Arman, and (**c**) Azad.

**Figure 3 foods-09-00113-f003:**
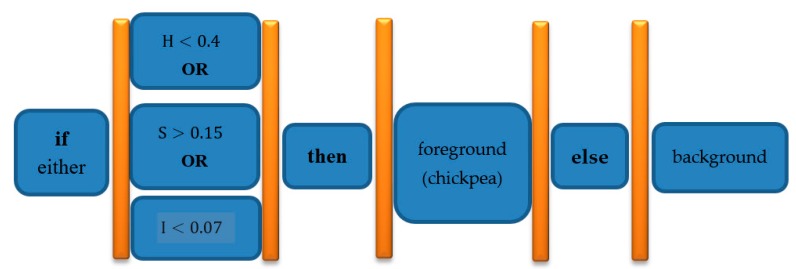
HSI color space segmentation Equation (1) flowchart for clarification purposes: background and foreground (chickpea) pixel segmentation.

**Figure 4 foods-09-00113-f004:**
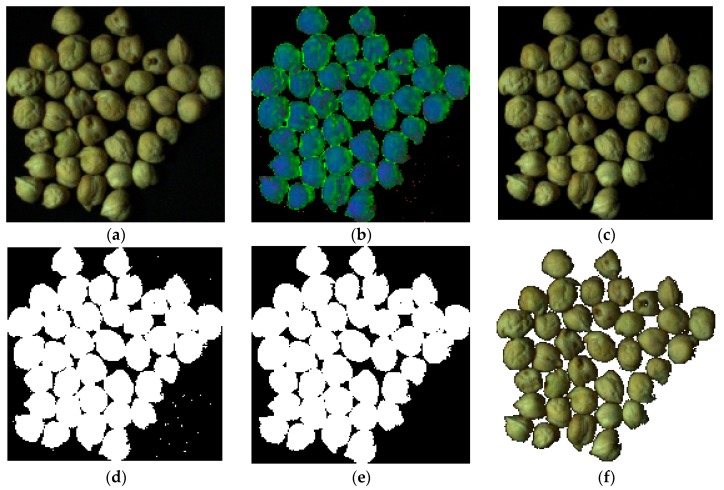
An example in the chickpea (*Cicer arietinum* L.) image segmentation process. (**a**): Original image, (**b**): Image converted to HSI color space, (**c**): Image after application of Equation (1), (**d**): Binary image, (**e**): Improved binary after remove objects with pixels less than 100 (this value select with try and error), (**f**): Segmented image.

**Figure 5 foods-09-00113-f005:**
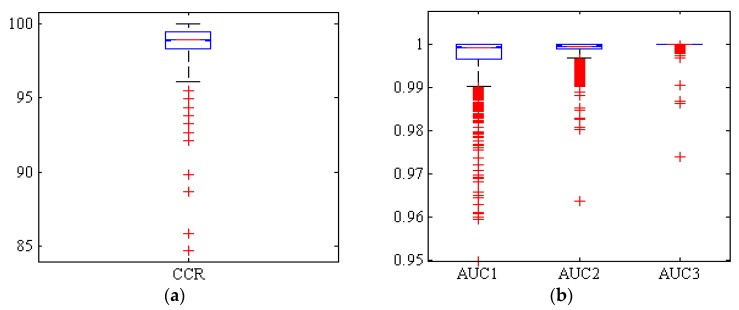
Boxplots of (**a**): the correct classification rate (CCR) and (**b**): area under the ROC curve (AUC) by hybrid ANN-PSO classifier, for Adel (AUC1), Arman (AUC2), and Azad (AUC3) chickpea (*Cicer arietinum* L.) varieties (1000 uniform random iterations, test set).

**Figure 6 foods-09-00113-f006:**
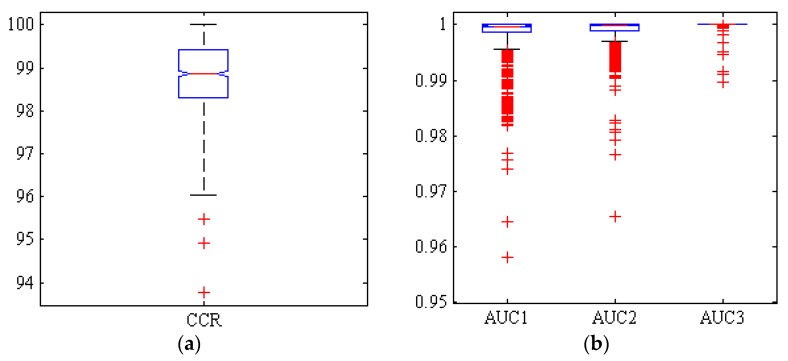
Boxplots of (**a**): the correct classification rate (CCR) and (**b**): area under the ROC curve (AUC) by hybrid ANN-CO classifier, for Adel (AUC1), Arman (AUC2) and Azad (AUC3) chickpea (*Cicer arietinum* L.) varieties (1000 uniform random iterations, test set).

**Figure 7 foods-09-00113-f007:**
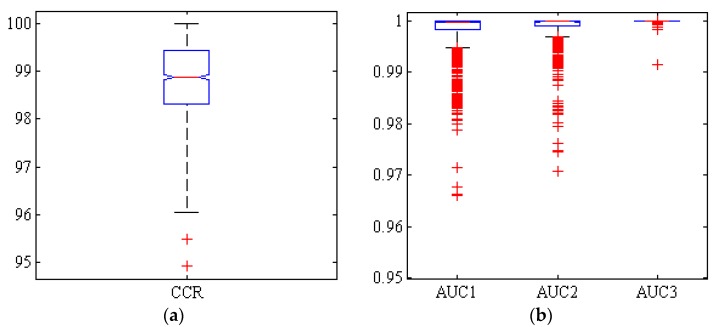
Boxplots of (**a**): the correct classification rate (CCR) and (**b**): area under the ROC curve (AUC) by hybrid ANN-HS classifier, for Adel (AUC1), Arman (AUC2) and Azad (AUC3) chickpea (*Cicer arietinum* L.) varieties (1000 uniform random iterations, test set).

**Figure 8 foods-09-00113-f008:**
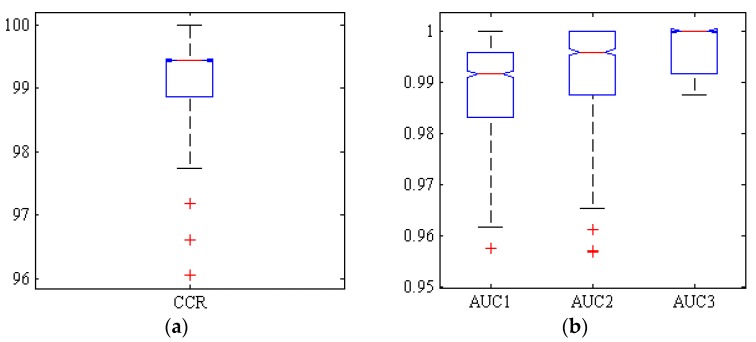
Boxplots of (**a**): the correct classification rate (CCR) and (**b**): AUC reached by ensemble ANN-PSO/ACO/HS-MV classifier, for Adel (AUC1), Arman (AUC2) and Azad (AUC3) chickpea (*Cicer arietinum* L.) varieties (1000 uniform random simulations, test set).

**Figure 9 foods-09-00113-f009:**
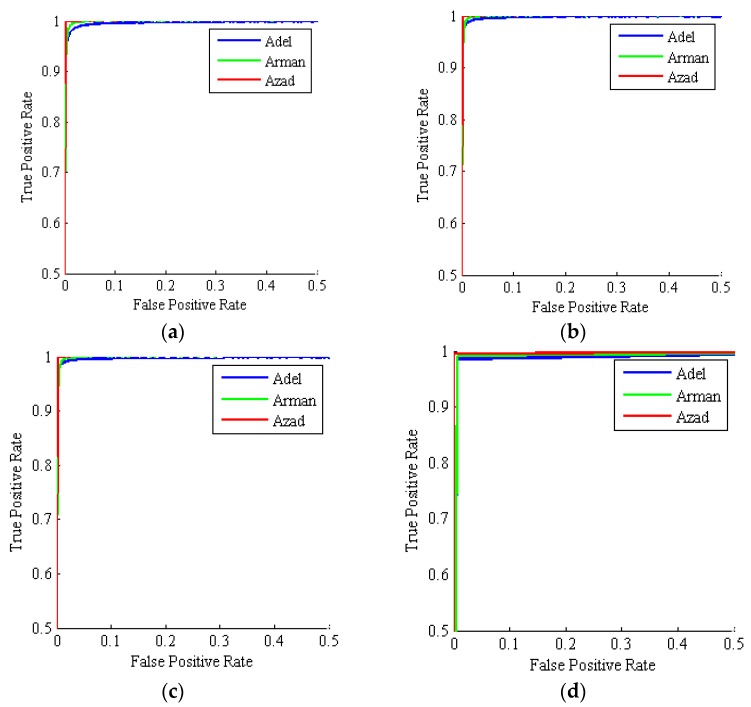
Mean ROC curves for three chickpea varieties: Adel (blue), Arman (green), Azad (red). (**a**) ANN-PSO; (**b**) ANN-ACO; (**c**) ANN-HS; (**d**) ANN-PSO/ACO/HS Majority-Voting ensemble (1000 simulations, test set).

**Table 1 foods-09-00113-t001:** Definition of the various color spaces used in the present work. Equations to obtain these channels from the original RGB values are also provided.

Color Space	Color Channel	Transformation from RGB Color Space
	V	V=M;withM=max{R,G,B};m=min{R,G,B};p=60 m/M
HSV	S	S=(M−m)/M
	H	H={p(G−B)ifM=R;120+p(B−R)ifM=G; 240+p(R−G)ifM=B}
HSI	I	I=(R+G+B)/3
	S	S=255−m/I
YCrCb	Cr	Cr=0.713(R−Y)+128
	Cb	Cb=0.564(B−Y)+128
YIQ	I	I=−0.595716·R−0.274453·G−0.321263·B
	Q	Q=0.211456·R−0.522581·G−0.311135·B
CMY	C	C=255−R
	M	M=255−G
	Y	Y=255−B

**Table 2 foods-09-00113-t002:** Texture features extracted from the gray level co-occurrence matrix (GLCM) [17].

Number	Feature Name	Number	Feature Name
1	Contrast	11	Inverse difference normalized (INN)
2	Sum of squares	12	Inverse difference moment normalized
3	Second diagonal moment	13	Diagonal moment
4	Mean	14	Sum average
5	Sum entropy	15	Variance
6	Difference variance	16	Sum variance
7	Difference entropy	17	Standard deviation
8	Information measure of correlation 1	18	Coefficient of variation
9	Information measure of correlation 2	19	Maximum probability
10	Inverse difference (INV) is homogeneity	20	Correlation

**Table 3 foods-09-00113-t003:** Parameter values in the artificial neural network (ANN) that are used in hybrid ANN-CA architecture in order to select the most effective (discriminant) features [19].

ANN Parameter	Value
Number of hidden layers	2
Number of neurons of the hidden layer	8, 19
Transfer function	*tribas, tansig*
Backpropagation network training function	*trainlm*
Backpropagation weight/bias learning function	*learncon*

**Table 4 foods-09-00113-t004:** Optimal architecture of hidden ANN layers determined by PSO, ACO and HS algorithms. [19]

Classifier	Num. of Layers	Number of Neurons	Transfer Function	Backpropagation Network Training Function	Backpropagation Weight/Bias Learning Function
ANN-PSO	3	First layer: 16	First layer: *netinv*	*learnlv1*	*traingdx*
		Second layer: 9	Second layer: *satlins*		
		Third layer: 18	Third layer: *compet*		
ANN-ACO	3	First layer: 12	First layer: *satlin*	*learnlv2*	*traingd*
		Second layer: 3	Second layer: *satlin*		
		Third layer: 13	Third layer: *poslin*		
ANN-HS	3	First layer: 13	First layer: *tansig*	*learnp*	*trainlm*
		Second layer: 10	Second layer: *satlin*		
		Third layer: 17	Third layer: *logsig*		

**Table 5 foods-09-00113-t005:** Confusion matrix including correct classification rate (CCR) for the ANN-PSO classifier: Adel, Arman and Azad chickpea (*Cicer arietinum* L.) varieties (1000 random iterations, test, train and validation sets).

Classifier	Data Set Type	Real/Estimated Class	Adel (1)	Arman (2)	Azad (3)	Total Data	Classification Error Per Class (%)	CCR (%)
ANN-PSO	Test	Adel	**57,854**	1048	98	59,000	1.94	98.65
		Arman	852	**57,147**	1	58,000	1.47	
		Azad	386	0	**59,614**	60,000	0.643	
	Train	Adel	**114,418**	3582	0	118,000	3.03	98.71
		Arman	0	**115,000**	0	115,000	0	
		Azad	936	0	**117,064**	118,000	0.793	
	Validation	Adel	**18,721**	272	7	19,000	1.47	98.09
		Arman	68	**18,932**	0	19,000	0.358	
		Azad	741	0	**18,259**	19,000	3.9	

**Table 6 foods-09-00113-t006:** Confusion matrix including correct classification rate (CCR) for ANN-ACO classifier: Adel, Arman and Azad chickpea (*Cicer arietinum* L.) varieties (1000 uniform random iterations, test train and validation sets).

Classifier	Data Set Type	Real/Estimated Class	Adel (1)	Arman (2)	Azad (3)	Total Data	Classification Error Per Class (%)	CCR (%)
ANN-ACO	Test	Adel	**58,059**	895	46	59,000	1.59	98.94
		Arman	656	**57,315**	29	58,000	1.18	
		Azad	243	0	**59,757**	60,000	0.405	
	Train	Adel	**117,074**	926	0	118,000	0.785	99.52
		Arman	753	**114,247**	0	115,000	0.655	
		Azad	0	0	**118,000**	118,000	0	
	Validation	Adel	**18,847**	153	0	19,000	0.805	98.88
		Arman	0	**18,804**	196	19,000	1.03	
		Azad	0	289	**18,711**	19,000	1.52	

**Table 7 foods-09-00113-t007:** Confusion matrix including correct classification rate (CCR) for ANN-HS classifier: Adel, Arman and Azad chickpea (*Cicer arietinum* L.) varieties (1000 uniform random iterations, test train and validation sets).

Classifier	Data Set Type	Real/Estimated Class	Adel (1)	Arman (2)	Azad (3)	Total Data	Classification Error Per Class (%)	CCR (%)
ANN-HS	Test	Adel	**58,059**	895	46	59,000	1.59	98.99
		Arman	601	**57,381**	18	58,000	1.07	
		Azad	235	0	**59,765**	60,000	0.392	
	Train	Adel	**116,841**	1159	0	118,000	0.982	99.67
		Arman	0	**115,000**	0	115,000	0	
		Azad	0	0	**118,000**	118,000	0	
	Validation	Adel	**18,841**	159	0	19,000	0.837	99.56
		Arman	89	**18,911**	0	19,000	0.468	
		Azad	0	0	**19,000**	19,000	0	

**Table 8 foods-09-00113-t008:** Confusion matrix including correct classification rate (CCR) for ensemble ANN Majority-Voting (MV): Adel, Arman and Azad chickpea (*Cicer arietinum* L.) varieties (1000 uniform random iterations, test set).

Ensemble Classifier	Real/Estimated Class	Adel (1)	Arman (2)	Azad (3)	Total Data	Classification Error Per Class (%)	CCR (%)
PSO/ACO/HS ensemble Majority-Voting	Adel	**58,184**	804	12	59,000	1.38	99.10
	Arman	508	**57,490**	2	58,000	0.879	
	Azad	266	0	**59,734**	60,000	0.443	

**Table 9 foods-09-00113-t009:** Comparison of the performance of the four classifiers (ANN-PSO, ANN-ACO, ANN-HS and ensemble ANN PSO/ACO/HS-MV): recall, specificity, precision, f1-score, AUC and accuracy (1000 iterations, test set).

Classifier	Class	Recall (%)	Specificity (%)	Precision (%)	F1-Score (%)	AUC (Mean ± Std. Dev.)	Accuracy % (Mean ± Std. Dev.)
ANN-PSO	Adel	97.91	99.03	98.06	97.98	0.9963 ± 0.0097	98.65 ± 1.31
	Arman	98.19	99.28	98.53	98.36	0.9988 ± 0.0026	
	Azad	99.83	99.66	99.36	99.59	0.9999 ± 0.0011	
ANN-ACO	Adel	98.47	99.2	98.4	98.44	0.9978 ± 0.0047	98.94 ± 0.89
	Arman	98.46	99.42	98.82	98.64	0.9888 ± 0.0029	
	Azad	99.87	99.79	99.59	99.73	0.9999 ± 0.0006	
ANN-HS	Adel	98.58	99.2	98.4	98.49	0.9975 ± 0.0051	98.99 ± 0.87
	Arman	98.46	99.48	98.93	98.69	0.9984 ± 0.0035	
	Azad	99.89	99.79	99.61	99.75	1.0000 ± 0.0004	
ensemble ANN PSO/ACO/HS Majority-Voting	Adel	98.69	99.31	98.62	98.65	0.9898 ± 0.0088	99.10 ± 0.75
	Arman	98.62	99.57	99.12	98.87	0.9822 ± 0.0083	
	Azad	99.97	99.77	99.56	99.77	0.9977 ± 0.0037	

**Table 10 foods-09-00113-t010:** Mean ± std. AUC values for the precision-recall curves(pr-AUC): ANN-PSO, ANN-ACO, ANN-HS and ensemble ANN PSO/ACO/HS-MV classifiers (1000 iterations, test set).

Chickpea Variety/Classifier	Adel	Arman	Azad
ANN-PSO	0.9763 ± 0.0206	0.9794 ± 0.0098	0.9822 ± 0.0295
ANN-ACO	0.9799 ± 0.0083	0.9796 ± 0.0085	0.9831 ± 0.0031
ANN-HS	0.9795 ± 0.0081	0.9775 ± 0.0132	0.9832 ± 0.0024
ensemble ANN PSO/ACO/HS Majority-Voting	0.9766 ± 0.1001	0.9756 ± 0.0122	0.9818 ± 0.0029

**Table 11 foods-09-00113-t011:** Accuracy of the classifiers reported in different food seeds. Please note no direct comparison possible given the various works do not share the same image database and are for different food seeds.

Paper	Number of Seed Samples	Correct Classification Rate (%)
Li et al. [27] (corn)	100	96.67
Men [28] (pea)	120	95
Aznan et al. [29] (rice)	120	96
Kurtulmus et al. [30] (pepper)	832	84.94
Here proposed (chickpea)	177,000	99.10

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
