# Peer review of "A Computer Vision System Based on Majority-Voting Ensemble Neural Network for the Automatic Classification of Three Chickpea Varieties"

_foods, 2020, doi:10.3390/foods9020113_

Round 1

Reviewer 1 Report

44) Why do the references start from 19? They are usually listed in order of quotation. This makes identification easier.

44) “ About 80-70%”   About 70-80%

44-46) It is not clear if the percentages refer to 100%

46)”[9]” Why do the reference is 9? They are usually listed in order of quotation. This makes identification easier.

46-51) It would be appropriate to explain how the industrial process of order preparation takes place. Are the chickpeas stored in the same tank once they are harvested? In this case the need for selection is evident. If this passage is well explained, the usefulness of the proposed methodology is highlighted.

53-54) You could list some of these methods. In this way the reader understands the difficulties in preparing the methodologies and maybe later on the usefulness of your methodology is highlighted.

54-55) Explain why these methods are low efficiency and capacity.

55-57) You should include cases of using these technologies (computer vision and machine learning), not only in these fields. To enrich the content of the paper you should present some cases of use of these technologies, briefly explaining how they are used to solve identification problems. There are many papers that use ML that you can mention. For example: "recursive partitioning for HVAC noise", “Random Forest Regression for Wind Turbine Noise Prediction”, and “Fault Diagnosis for UAV Blades”. Furthermore, for computer vision: “Feature extraction and image processing for computer vision”, and “Deep learning for computer vision: A brief review”.

61-62) “Mass level imaging vs kernel surface imaging” Explain what you are talking about. Readers may not know these technologies.

66) “GLCM-based, GLRLM-based and local binary patterns” Explain what you are talking about. Readers may not know these technologies. Don't use acronyms if you haven't used the full name before.

72) “SVM”. Don't use acronyms if you haven't used the full name before.

80-83) Properly introduce the purpose of the paper. Explain better the algorithms you will use and do not use acronyms in this phase.

85-87) The use of a diagram to summarize the flow of operations is advisable, but you should first introduce them properly step by step. It is a paper you have all the space available to explain what you are going to do.

92-93) Describe the three varieties in detail. Explain what the differences are, represent the object of your classification.

95)” 196, 192 and 197 images were obtained”. Describe in detail how the images were produced. How were the chickpeas arranged in the images? How many chickpeas were used in the images? How do images differ from each other?

107) “HSI” Use the long form name for first. Then you can use the acronym.

107-109) Introduce “color space” topic, and explain why this topic is crucial for addressing the objects identification.

109-111) If you had introduced the technology, the reference would now be clear.

112-113) Where does equation 1 come from? How was it processed?

114) Before introducing Figure 3 you should explain the procedure for segmentation step by step. This is an essential procedure in your work.

128-129)” RGB, HSV, HSI, YCbCr, CMY and YIQ color spaces”. Use the long form name and explain the terms.

129) Insert references for these technologies.

129) “four channels”. Define the 4 channels

131) “GLCM matrix” Explain the topic.

135) Summarize the number of features you've extracted.

139-144) Feature extraction is also important for the ability to generalize the model. Eliminating the redundancy of information helps the learning process.

144-145) “hybrid 144 ANN-CA”. Use the long form name and explain the term.

145) Introduce first the ANN and then the CA. Add references for those wishing to learn more about topics, such as books and papers.

145) “Cultural algorithm (CA)” Perfect, first long name and then the acronym.

150-153) Attention, repeat the same period twice.

154) “the artificial”. Between the words there is an extra space

154-155) “The artificial neural network divides all the data samples into three disjoint sets. ” In reality it is not the network but it is we who divide the dataset into three parts and then use them for training, validation and testing.

155-157) How is splitting done? Indicates the procedure.

159-162) Explain this phase of the study better. How you use the metric for evaluating the algorithm's performance.

165) In table 2 you quote different algorithms. You should briefly indicate what it is: tribas, tansig, trainlm, learncon. Insert also some references.

169) “ANN majority-voting (MV)”. Introduce adequately the topic.

173-176) Perfect, you could introduce these features when explaining the ANNs. Now that you've introduced them you could briefly explain what they are. Insert also some references.

178-183) Many abbreviations without any explanation, I believe that in this way it does not help the reader.

183) “Particle Swarm Optimization (PSO)”. Before introducing the algorithm, explain what you will use it for.

195) Move this sentence at the 183 line.

199-200) You could insert a chart with the results of the investigation.

205-212) Explain well the principle of this algorithm, you should explain just as well how you use this algorithm in your case.

222) “MV approach” You didn't introduce it, explain what it is in.

226) Introduce adequately Table 3. Explain what it shows add two lines to the methods used.

229) “confusion matrtices” . confusion matrices

239-247) You could mention a book from which you extracted these definitions (for example “Neural Networks with R”). In these definitions you have been very precise, you should do the same for the algorithms you mentioned in the previous paragraphs.

249-250) In equation 3 there is an extra +

257-267) You could insert a ROC chart to better understand its meaning

291) After Figure 4, explain what we can see in the boxplot.

312) After Figure 5, explain what we can see in the boxplot.

327) After Figure 6, explain what we can see in the boxplot.

362) After Figure 8, explain what we can see in the ROC curve.

373-387) In the Conclusions you should summarize what you did in this work. What were your goals and what results did you achieve? Finally you should specify what the practical uses of the technology you are proposing are.

The authors must enrich the bibliography. it is necessary to include more references to works that have used similar technologies. Furthermore, the algorithms used must be better explained and in particular how they were used for this work.

Author Response

Reviewer 1

44) Why do the references start from 19? They are usually listed in order of quotation. This makes identification easier.

A: Thank you for pointing out. Done.

44) “ About 80-70%”About 70-80%

A: Done.

44-46) It is not clear if the percentages refer to 100%

A: Done.

46)”[9]” Why do the reference is 9? They are usually listed in order of quotation. This makes identification easier.

A: Thank you for pointing out. Done.

46-51) It would be appropriate to explain how the industrial process of order preparation takes place. Are the chickpeas stored in the same tank once they are harvested? In this case the need for selection is evident. If this passage is well explained, the usefulness of the proposed methodology is highlighted.

A: Done. Following paragraph was added in revised manuscript:

One can consider at least two main reasons why it might be important to provide rapid detection systems using new technology: first, different applications and price of each chickpea variety; second, separation of each chickpea variety when stored in the same tank, once they have been harvested.

53-54) You could list some of these methods. In this way the reader understands the difficulties in preparing the methodologies and maybe later on the usefulness of your methodology is highlighted.

A: Done. The following paragraph was added to revised manuscript text:

For instance, in order to separate ripe fruit from immature, fruits were fed in a row on a v-shaped conveyor. At the sorting points, the piston behind the belt vibrated at a predetermined resonance frequency producing movement over fruits. Vibrant immature fruits were cut into the lateral conveyor belts and separated from ripened ones. Another example was the use of elasticity feature in sorting of ripe fruit. Fruits were given a free fall onto a rotating cylinder. As the rotary cylinder moved, the fruits had a specific downward parabola motion. Ripe fruit with soft texture would be directed to closer outlet and unripe fruit with firm texture would be directed farther to the second outlet. Since these classical food separation systems have a high response time as compared to new computer vision techniques, thus classical system capacity is low. In addition, it is also possible to damage fruit piece texture due to free fall or mechanical vibration [6]

[6] Carl W. Hall and Denny C. Davis. Processing Equipment for Agricultural Products. The AVI Publishing Inc, 2nd Edition, 1979.

54-55) Explain why these methods are low efficiency and capacity.

A: Done. The following paragraph was added to revised manuscript text:

For instance, in order to separate ripe fruit from immature, fruits were fed in a row on a v-shaped conveyor. At the sorting points, the piston behind the belt vibrated at a predetermined resonance frequency producing movement over fruits. Vibrant immature fruits were cut into the lateral conveyor belts and separated from ripened ones. Another example was the use of elasticity feature in sorting of ripe fruit. Fruits were given a free fall onto a rotating cylinder. As the rotary cylinder moved, the fruits had a specific downward parabola motion. Ripe fruit with soft texture would be directed to closer outlet and unripe fruit with firm texture would be directed farther to the second outlet. Since these classical food separation systems have a high response time as compared to new computer vision techniques, thus classical system capacity is low. In addition, it is also possible to damage fruit piece texture due to free fall or mechanical vibration [6]

55-57) You should include cases of using these technologies (computer vision and machine learning), not only in these fields. To enrich the content of the paper you should present some cases of use of these technologies, briefly explaining how they are used to solve identification problems. There are many papers that use ML that you can mention. For example: "recursive partitioning for HVAC noise", “Random Forest Regression for Wind Turbine Noise Prediction”, and “Fault Diagnosis for UAV Blades”. Furthermore, for computer vision: “Feature extraction and image processing for computer vision”, and “Deep learning for computer vision: A brief review”.

A: Done. The following paragraph was added to revised manuscript text:

Iannace et al. developed a model based on artificial neural network to detect unbalanced blades in an Unmanned Aerial Vehicle (UAV) propeller. An air collision or a defect in the propellers of a drone can cause the aircraft to fall to the ground and its consequent destruction [9]. LeCun et al. recognized document content using a Gradient-based learning rule [10]. Uijlings et al. introduce a selective search which combines the strength of both an exhaustive search and segmentation for object recognition that results in a small set of data-driven, class-independent, high quality locations, yielding a 99% recall value [11].

[9] Gino Iannace *, Giuseppe Ciaburro * and Amelia Trematerra, 2019. Fault Diagnosis for UAV Blades Using Artificial Neural Network. Robotics. 8 (59)

[10] Y. LeCun, L. Bottou, Y. Bengio, and P. Hafner, “Gradient-based learning applied to document recognition,” Proceedings of the IEEE, vol. 86, no. 11, pp. 2278–2323, 199

[11] J. R. R. Uijlings, K. E. A. Van De Sande, T. Gevers, and A. W. M. Smeulders, “Selective search for object recognition,” International Journal of Computer Vision, vol. 104, no. 2, pp. 154–171, 2013.

61-62) “Mass level imaging vs kernel surface imaging” Explain what you are talking about. Readers may not know these technologies.

A: done

66) “GLCM-based, GLRLM-based and local binary patterns” Explain what you are talking about. Readers may not know these technologies. Don't use acronyms if you haven't used the full name before.

A: done

72) “SVM”. Don't use acronyms if you haven't used the full name before.

A: SVM is the acronyms of Support Vector Machine. Done.

80-83) Properly introduce the purpose of the paper. Explain better the algorithms you will use and do not use acronyms in this phase.

A: Done. The following paragraph was added to revised manuscript text:

The aim of the present paper is to develop a computer vision system in order to classify three similar varieties of chickpea (Adel, Azad and Arman) using image processing and machine learning ensemble ANN Majority Voting (MV) approach between Hybrid Artificial Neural Network- Particle Swarm Optimization (ANN-PSO), Hybrid Artificial Neural Network-Ant Colony Optimization (ANN-ACO) and Hybrid Artificial Neural Network-Harmonic Search (ANN-HS) outputs.

85-87) The use of a diagram to summarize the flow of operations is advisable, but you should first introduce them properly step by step. It is a paper you have all the space available to explain what you are going to do.

A: Done. The following paragraph was added to revised manuscript text:

In this study, in order to train a computer vision system to be able to identify different chickpea varieties, different steps are required. Figure 1 illustrates a flowchart for the various stages needed to properly design and functioning of our system. As it can be seen in Fig. 1, five main blocks exist in our chickpea automatic classification system, being each block explained next.

92-93) Describe the three varieties in detail. Explain what the differences are, represent the object of your classification.

A: Done. The following paragraph was added to revised manuscript text:

Adel variety is used mainly for making cakes and muffins in the food industry, Arman variety is used for livestock consumption and Azad variety is used as powder in various food preparations.

95)” 196, 192 and 197 images were obtained”. Describe in detail how the images were produced. How were the chickpeas arranged in the images? How many chickpeas were used in the images? How do images differ from each other?

A: Done. The following paragraph was added to revised manuscript text:

Chickpeas are spread randomly, are not isolated and they might touch one to each other. Each image contains around 60 chickpea samples. After each capture, chickpeas were removed and replaced with a different batch sample

107) “HSI” Use the long form name for first. Then you can use the acronym.

A: Done.

107-109) Introduce “color space” topic, and explain why this topic is crucial for addressing the objects identification.

A: Done. The following paragraph was added to revised manuscript text:

As is well-known, a color space is an input space including three main colors components. Each color space presents a particular value of a vector size value for each pixel, comprising each of the three scalar color components in that particular color space. Since the values of pixels in background and chickpeas are different, it might be possible to separate pixels in chickpea (foreground) from pixels in the background.

112-113) Where does equation 1 come from? How was it processed?

A: It was set by trial and error, and used for segmentation purposes. A valid threshold needs to be defined for each color channels value in HSI color space. Equation (1) describes segmentation threshold in formal terms. This equation indicates that: if the pixel with the first Hue channel of the color space is less than 0.4 or the second Saturation channel of this color space is greater than 0.15 or the third channel Intensity of this color space is greater than 0.07, then the pixel belongs to chickpea (foreground), otherwise pixel is part of background.

114) Before introducing Figure 3 you should explain the procedure for segmentation step by step. This is an essential procedure in your work.

A: Done. The following paragraph was added to revised manuscript text:

Figure 3 shows a sample image through the whole segmentation process. As first step, the original image is convert to HSI color space; in second step, based on equation (1), most background pixels are remove; finally remain background pixels are remove based on morphology operator, removing objects with a number of pixels of less than 100. Last value was select after trial and error.

128-129)” RGB, HSV, HSI, YCbCr, CMY and YIQ color spaces”. Use the long form name and explain the terms.

A: Done. The following new table was added to revised manuscript text:

Table 1. Definition of the various color spaces used in the present work. Equations to obtain these channels from the original RGB values are also provided.

Color space

Color Channel

Transformation from RGB color space

HSV

HSI

YCrCb

YIQ

CMY

129) Insert references for these technologies.

A: Done, the following reference was added: (Pourdarbani et al., 2019).

[16] Razieh Pourdarbani, Sajad Sabzi, Mario Hernández-Hernández, José Luis Hernández-Hernández, Ginés García-Mateos, Davood Kalantari, and José Miguel Molina-Martínez. Comparison of Dierent Classifiers and the Majority Voting Rule for the Detection of Plum Fruits in Garden Conditions. Remote Sens. 2019, 11, 2546; doi:10.3390/rs11212546

129) “four channels”. Define the 4 channels

A: True, it was not clear before. We added the following text to revised manuscript:

Since there are two properties, four channels (three color components plus the average of all them three) and six color spaces, the total number of extractable properties in this case equals .

131) “GLCM matrix” Explain the topic.

A: This is the well-known classical Gray Level Co-occurrence Matrix (GLCM) method for computing texture features.

135) Summarize the number of features you've extracted.

A: Sorry but we ca not summarize here the number of features, since we used those features in the effective feature selection stage.

144-145) “hybrid 144 ANN-CA”. Use the long form name and explain the term.

A: Done. Hybrid Artificial Neural Network-Cultural Algorithm (ANN-CA).

145) Introduce first the ANN and then the CA. Add references for those wishing to learn more about topics, such as books and papers.

A: In this section the focus is over the CA algorithm, In Section 2.5. we explained about various ANNs. For CA algorithm we already cited a reference, and for ANNs we add new reference:

Caudill, M., Neural Networks Primer, San Francisco, CA: Miller Freeman Publication, 1989.

145) “Cultural algorithm (CA)” Perfect, first long name and then the acronym.

150-153) Attention, repeat the same period twice.

A: True, thank you, we removed it.

154) “the artificial”. Between the words there is an extra space

A: True, thank you. It was corrected.

154-155) “The artificial neural network divides all the data samples into three disjoint sets. ” In reality it is not the network but it is we who divide the dataset into three parts and then use them for training, validation and testing.

A: Ok. It was corrected as follows:

Data samples were randomly divided into three disjoint sets.

155-157) How is splitting done? Indicates the procedure.

A: Data were randomly divided into three disjoint sets. using code written in MatLab software, as was explained in manuscript text:

The first set contains 70% of the data samples for training, the second one contains 15% of the data for validation and the third one contains 15% of the data for testing, being all sets disjoint

159-162) Explain this phase of the study better. How you use the metric for evaluating the algorithm's performance.

A: Mean Square Error (MSE) is a measure used to evaluate the performance of ANN. An MSE value close to 0 implies that the ANN have learned very good. For this reason, in that section MSE was used in order to select the best vecyor of adjustable parameters.

165) In table 2 you quote different algorithms. You should briefly indicate what it is: tribas, tansig, trainlm, learncon. Insert also some references.

A: tribas, tansig are transfer functions, trainlm is backpropagation network training function and learncon is backpropagation weight/bias learning function. Additional help and explanation of those functions can be read in MatLab software manuals.

169) “ANN majority-voting (MV)”. Introduce adequately the topic.

A: Done. The following text was added to revised manuscript text:

MV decision is based on voting from each independent classifier and agreeing as final consensus winning output the class which was voted most among the various single classifiers.

173-176) Perfect, you could introduce these features when explaining the ANNs. Now that you've introduced them you could briefly explain what they are. Insert also some references.

A: Done. New referenced inserted:

Caudill, M., Neural Networks Primer, San Francisco, CA: Miller Freeman Publication, 1989.

178-183) Many abbreviations without any explanation, I believe that in this way it does not help the reader.

A: Please note that those abbreviations are well-known functions properly defined in MatLab software.

183) “Particle Swarm Optimization (PSO)”. Before introducing the algorithm, explain what you will use it for.

A: In this section, Particle Swarm Optimization (PSO) was used for adjusting these adjustable parameters, in an optimal way.

205-212) Explain well the principle of this algorithm, you should explain just as well how you use this algorithm in your case.

A: The steps of selecting the ANN- ACO adjustable parameters are exactly like the PSO algorithm that used in hybrid ANN-PSO.

226) Introduce adequately Table 3. Explain what it shows add two lines to the methods used.

A: OK, done. We added the following sentence to revised manuscript text:

As it can be inferred from Table 4, the best structure for all methods is a structure with three layers. Also this table we show that the best number of neurons of each layer that was selected based on the ACO, PSO and HS algorithms, are below 20 neurons thus implying that there is no need for networks to be highly complex.

229) “confusion matrtices” . confusion matrices

A: True, corrected. Thank you.

239-247) You could mention a book from which you extracted these definitions (for example “Neural Networks with R”). In these definitions you have been very precise, you should do the same for the algorithms you mentioned in the previous paragraphs.

A: Done:

Wisaeng, K. 2013. A comparison of decision tree algorithms for UCI repository classification. Int. J. Eng. Trends Technol. 4, 3393-3397.

249-250) In equation 3 there is an extra +

A: True, thank you. Corrected.

257-267) You could insert a ROC chart to better understand its meaning

A: Done, we added it in the Discussion section.

291) After Figure 4, explain what we can see in the boxplot.

A: It was already explained in manuscript text, but before Figure 4:

Figure 4 shows a box diagram of CCR and AUC levels below the ROC curve of Adel, Arman and Azad classes (AUC1, AUC2, and AUC3, respectively). Among the 1,000 replicates, only 10 replicates had a CCR below 95%. Also, more than 50% of replicates have a CCR above 99%. Therefore, it can be concluded that this classifier has been able to be performed the classification with high accuracy.

312) After Figure 5, explain what we can see in the boxplot.

A: It was already shown in original version of manuscript:

Figure 5 illustrates a box diagram of CCR and AUC under ROC curve of the three classes for ANN-ACO classifier, 1,000 random repeated simulations. As can be seen from Figure 5, in this case only two simulations out of 1,000 have a correct classification rate below 95%. On the other hand, the boxplot diagrams for the areas under the ROC curve (AUC) of the three chickpea varieties are presented, with values above 0.95 in all 1,000 simulations. The more compact box diagram however, indicates the closer proximity of the results in different iterations, resulting in a high level of classification accuracy with very limited dispersion.

327) After Figure 6, explain what we can see in the boxplot.

A: It was already explained in manuscript text, but before Figure 6:

Figure 6 shows the box diagram of the CCR, and AUC of hybrid ANN-HS after 1,000 iterations. It also shows that more than half of the simulations have a correct classification rate over 99%.

362) After Figure 8, explain what we can see in the ROC curve.

A: It was already explained in manuscript text, but before Figure 8:

Figure 8 shows the ROC curves by the four hybrid classifiers: ANN-PSO, ANN-ACO, ANN-HS and ANN-PSO/ACO/HS ensemble Majority-Voting (MV). As can be seen, the ROC curves for each of the three classes produced by the ANN-MV scheme are very close to the top-left chart corner, indicating its superior performance.

373-387) In the Conclusions you should summarize what you did in this work. What were your goals and what results did you achieve? Finally you should specify what the practical uses of the technology you are proposing are.

A: We extended the Conclusions section, avoiding repetitions from Discussion Section, with the following text:

Given the high accuracy of the approach here propose, we believe it could in principle be extended and applied under real conditions in the food industry in order to automatically classify several chickpea varieties.

The authors must enrich the bibliography. it is necessary to include more references to works that have used similar technologies.

A: Done. We added some new references in text, highlighted.

Furthermore, the algorithms used must be better explained and in particular how they were used for this work.

A: OK, done. We add new text at the end of the Discussion Section:

To sum up, the proposed system works in four different stages described as follows:

Chickpea bunch imaging acquisition under light controlled conditions, with white LEDs with 425 lux intensity. Automatic chickpea image segmentation. Automatic extraction of different discriminant features, including: average channels of first, second and third RGB color space, average first channel of HSI color space, neighborhood Entropy of 90°and 0° in GLCM, and third channel of YCbCr color space, from each input sample image. Output chickpea variety classification by a neural network ensemble Majority-Voting.

Reviewer 2 Report

Not sure whether the author split the data into training, test, and validation. The reviewer would like to see the confusion matrix for each.

The sample size for each class should be clearly mentioned. The reviewer cant find how many the images the authors have for each class.

Sensitivity study to find the number of neurons should be presented.

Author Response

Reviewer 2

Not sure whether the author split the data into training, test, and validation. The reviewer would like to see the confusion matrix for each.

The sample size for each class should be clearly mentioned. The reviewer cant find how many the images the authors have for each class.

A: 60 % of images from each class were used for training, 10 % of them were used for validation and 30 % of them were used for testing, disjoint sets. For example, Adel class that has 196 images, thus 196*0.6=118 images were used for training, 196*0.1=19 images were used for validation and 196*0.3=59 images were used for test set.

In addition, as requested by reviewer, we added confusion matrix for both train and validation sets, in new Tables 5, 6 and 7, as shown next, despite the really important one are those for test set, for obvious reasons (test set has input samples never seen before by the ANNs while in the learning phase), which should be the one used for performance evaluation and further comparison:

Table 5. Confusion matrix including correct classification rate (CCR) for the ANN-PSO classifier: Adel, Arman and Azad chickpea (Cicer arietinum L) varieties (1,000 uniform random iterations, test, train and validation sets).

Classifier

data set type

real / estimated class

Adel (1)

Arman (2)

Azad (3)

Total data

Classification error per class (%)

CCR (%)

ANN-PSO

Test

Adel

57854

1048

98

59000

1.94

98.65

Arman

852

57147

1

58000

1.47

Azad

386

0

59614

60000

0.643

Train

Adel

114418

3582

0

118000

3.03

98.71

Arman

0

115000

0

115000

0

Azad

936

0

117064

118000

0.793

Validation

Adel

18721

272

7

19000

1.47

98.09

Arman

68

18932

0

19000

0.358

Azad

741

0

18259

19000

3.9

Table 6. Confusion matrix including correct classification rate (CCR) for ANN-ACO classifier: Adel, Arman and Azad chickpea (Cicer arietinum L) varieties (1,000 uniform random iterations, test train and validation sets).

Classifier

data set type

real / estimated class

Adel (1)

Arman (2)

Azad (3)

Total data

Classification error per class (%)

CCR (%)

ANN-ACO

Test

Adel

58059

895

46

59000

1.59

98.94

Arman

656

57315

29

58000

1.18

Azad

243

0

59757

60000

0.405

Train

Adel

117074

926

0

118000

0.785

99.52

Arman

753

114247

0

115000

0.655

Azad

0

0

118000

118000

0

Validation

Adel

18847

153

0

19000

0.805

98.88

Arman

0

18804

196

19000

1.03

Azad

0

289

18711

19000

1.52

Table 7. Confusion matrix including correct classification rate (CCR) for ANN-HS classifier: Adel, Arman and Azad chickpea (Cicer arietinum L) varieties (1,000 uniform random iterations, test train and validation sets).

Classifier

data set type

real / estimated class

Adel (1)

Arman (2)

Azad (3)

Total data

Classification error per class (%)

CCR (%)

ANN-HS

Test

Adel

58059

895

46

59000

1.59

98.99

Arman

601

57381

18

58000

1.07

Azad

235

0

59765

60000

0.392

Train

Adel

116841

1159

0

118000

0.982

99.67

Arman

0

115000

0

115000

0

Azad

0

0

118000

118000

0

Validation

Adel

18841

159

0

19000

0.837

99.56

Arman

89

18911

0

19000

0.468

Azad

0

0

19000

19000

0

Sensitivity study to find the number of neurons should be presented.

A: We used different algorithm such as ACO, PSO and HS for selecting the best value not only for the number of neurons but also for all five adjustable parameters in neural network, including: the number of layers, the transfer function, the back-propagation network training function, and the back-propagation weight / bias learning function, what supplements the sensitivity study requested by reviewer.

Round 2

Reviewer 1 Report

59-68) Comprehensive explanation, everything is now clearer.

150) Equation 1 could be proposed as a flow chart to make understanding more immediate

236) In table 3 you quote different algorithms. You should briefly indicate what it is: tribas, tansig, trainlm, learncon. Insert also some references.

“A: tribas, tansig are transfer functions, trainlm is backpropagation network training function and learncon is backpropagation weight/bias learning function. Additional help and explanation of those functions can be read in MatLab software manuals.”

The authors' answer is acceptable, however there is no reference to a Matlab resource where to retrieve this information. Insert a reference to a Matlab manual where these information are reported. There are many books that explain this information.

256-258)” Theese abreviations are 256 functions defined under MatlLab software.” Insert a reference for these abbreviations. Where the readers can find an adequate explanation of the meaning?

242-244) Insert a reference to MV method.

Author Response

Reviewer 1:

59-68) Comprehensive explanation, everything is now clearer.

A: We are glad, thank you.

150) Equation 1 could be proposed as a flow chart to make understanding more immediate

 A: Done, as shown next:

if either then foreground (chickpea), else background

(1)

Figure 3 shows a flowchart to better understand the meaning of segmentation equation (1).

Figure 3. HSI color space segmentation equation (1) flowchart for clarification purposes: background and foreground (chickpea) pixel segmentation.

236) In table 3 you quote different algorithms. You should briefly indicate what it is: tribas, tansig, trainlm, learncon. Insert also some references.

“A: tribas, tansig are transfer functions, trainlm is backpropagation network training function and learncon is backpropagation weight/bias learning function. Additional help and explanation of those functions can be read in MatLab software manuals.”

The authors' answer is acceptable, however there is no reference to a Matlab resource where to retrieve this information. Insert a reference to a Matlab manual where these information are reported. There are many books that explain this information.

A: True, done, following referenced added inside revised manuscript text. Thank you.

N. Sivanandam, S. N Deepa. Introduction to Neural Networks Using Matlab 6.0. Tata McGraw-Hill Education, 2006. ISBN: 0070591121, 9780070591127.

256-258)” Theese abreviations are 256 functions defined under MatlLab software.” Insert a reference for these abbreviations. Where the readers can find an adequate explanation of the meaning?

A: True, done, following referenced added inside revised manuscript text. Thank you.

N. Sivanandam, S. N Deepa. Introduction to Neural Networks Using Matlab 6.0. Tata McGraw-Hill Education, 2006. ISBN: 0070591121, 9780070591127.

242-244) Insert a reference to MV method.

A: True, done, following referenced added inside revised manuscript text. Thank you.

Grzegorz Zabinski, Jarosław Gramacki, Artur Gramacki, Ewelina Mista-Jakubowska, Thomas Birch, Alexandre Disser. Multi-classifier majority voting analyses in provenance studies on iron artefacts. Journal of Archaeological Science 113 (2020) 105055. https://doi.org/10.1016/j.jas.2019.105055

Reviewer 2 Report

The reviewers have satisfactorily addressed the comments.

Author Response

Reviewer 2:

The reviewers have satisfactorily addressed the comments

A: Thank you.
